# Assessing the Volume of the Head of the Mandibular Condyle Using 3T-MRI—A Preliminary Trial

**DOI:** 10.3390/dj12070220

**Published:** 2024-07-16

**Authors:** Alessandro Mosca Balma, Davide Cavagnetto, Lorenzo Pavone, Federico Mussano

**Affiliations:** 1Bone and Dental Bioengineering Laboratory, CIR Dental School, Department of Surgical Sciences, University of Turin, Via Nizza 230, 10126 Turin, Italy; alessandro.moscabalma@unito.it (A.M.B.); lorenzo.pavone@unito.it (L.P.); federico.mussano@unito.it (F.M.); 2Department of Mechanical and Aerospace Engineering, Polytechnic of Turin, Corso Duca degli Abruzzi 24, 10129 Turin, Italy

**Keywords:** magnetic resonance imaging, three-dimensional imaging, cone beam computed tomography, digital dentistry, orthodontics, semiautomatic volumetric segmentation

## Abstract

Due to potentially harmful exposure to X-rays, condylar growth in response to orthodontic treatment is poorly studied. To overcome this limitation, here, the authors have proposed high-resolution MRI as a viable alternative to CBCT for clinical 3D assessment of TMJ. A male subject underwent both MRI and CBCT scans. The obtained three-dimensional reconstructions of the TMJ were segmented and superimposed by a semiautomatic algorithm developed in MATLAB R2022a. The condylar geometries were reconstructed using dedicated software for image segmentation. Two geometrical parameters, i.e., the total volume and surface of the single condyle model, were selected to quantify the intraclass and interclass variability from the mean of each DICOM series (CBCT and MRI). The final comparison between the reference standard model of CBCT and 3T MRI showed that the former was more robust in terms of reproducibility, while the latter reached a higher standard deviation compared to CBCT, but these values were similar between the operators and clinically not significant. Within the inherent limitation of image reconstruction on MRI scans due to the current lower resolution of this technique, the method proposed here could be considered as a nucleus for developing future completely automatic AI algorithms, owing to its great potential and satisfactory consistency among different times and operators.

## 1. Introduction

It is widespread knowledge that computed tomography (CT) offers clearer depictions of internal bone structures and calcifications compared to conventional MR imaging. Additionally, its benefits encompass shorter examination time, relatively economical expenses, and convenient accessibility, rendering CT the preferred technique for bone imaging over an extended period. The portrayal of solid bone structures via MRI poses a challenge due to their minimal proton density (approximately 20% of water) and extremely brief T2 relaxation time (around 390 μs at 3 T) [1,2,3]. Despite the emergence of MR bone imaging (resembling CT) utilizing a brief echo time (TE) as a novel technology in recent times, its recognition remains limited [2,3]. Nonetheless, unlike CT, which necessitates exposure to ionizing radiation, MR bone imaging holds potential for examining various regions, including bones and adjacent soft tissue structures like ligaments.

Several fields of dentistry could benefit from this 3D radiation-free imaging technique from clinical and research perspectives. In particular, orthodontics research has recently been focused on several unanswered questions, mostly due to the inherent limitation of the bidimensional conventional imaging routinely used (i.e., cephalometric analysis) and the radiation exposure burden connected to 3D CT analysis.

Cephalometric analysis (CA) is the assessment of the spatial relationships among bones and teeth as attained through the calculation of angular and linear measurements on anatomic landmarks based on craniofacial radiographic images [4,5]. In maxillofacial surgery and orthodontics, CA is the method of choice for diagnosing craniofacial anomalies, and planning and evaluating treatment outcomes. So far, two-dimensional (2D) radiographs have been used preferably for CA, even leading some authors to propose the synthesis of lateral cephalograms from cone beam computed tomography (CBCT) images to perform conventional 2D CA [6,7,8]. This approach seems hindered, however, by limitations such as geometric distortions and superimpositions [6,9]. 

Since cone beam computed tomography (CBCT) has become the standard three-dimensional (3D) imaging technique in dentistry [10,11], 3D cephalometry performed on CBCT has acquired growing interest due to its excellent spatial resolution (250 μm isotropic voxel size or lower) and high geometric accuracy [4]. Compared to 2D radiographs, CBCT provides substantial diagnostic benefits for CA, as regards an overall enhanced accuracy and the improved detection and quantification of craniomaxillofacial asymmetries [7,8,9,12]. Unfortunately, however, these advantages come at the expense of an augmented X-ray dose [13]. Moreover, radiation risks in CBCT increase with larger fields of view (FOVs) and lower age [14]. This must be taken into consideration when dealing with 3D cephalometry that requires large scanning volumes and is administered to patients who are often adolescent or younger [14]. Especially in these patient groups, radiation awareness and safety have the utmost importance according to the principles of the “Image Gently” campaign [15].

Magnetic resonance imaging (MRI) has emerged as a potential non-ionizing alternative in the 3D assessment of the maxillofacial structures [16,17,18]. In principle, MRI allows the detection of both soft and hard oral tissues—including tooth surfaces—enabling their measurement [16,19]. Indeed, higher resolution and reduction in susceptibility artifacts have been achieved recently thanks to dedicated coil and sequence techniques that allow us to achieve results of volumetric assessment similar to CBCT scans [20]. A remarkable study in vivo reported excellent geometric accuracy and high reproducibility of 3D cephalometric measurements performed on MRI [19], supporting the potential of MRI to provide 3D information comparable to CBCT, thus overcoming the radiation dose dilemma, particularly in young patients.

Condylar changes are of paramount importance in orthodontic treatment planning, but no clear evidence has yet been provided on condylar growth in response to functional stimuli. Here, the authors aim to test whether MRI could be as reliable a technique as CBCT for 3D assessment in vivo of the mandibular condyles. According to the authors’ knowledge, this is the first attempt to develop a reliable MRI-based method to acquire volumetric information regarding maxillofacial structures recurring in a semiautomatic algorithm for clinical and research purposes.

## 2. Materials and Methods

### 2.1. Case Description

A male subject aged 30 underwent CBCT and 3T MRI scans after a car accident as follow-up examinations.

### 2.2. CBCT and MRI Acquisitions

The CBCT scan was taken using a cone beam i-CAT FLX unit (Imaging Sciences International, Inc., Hatfield, PA, USA. https://ct-dent.co.uk/i-cat-vision/ (accessed on 4 April 2024)). The machine was set for full rotation, at 300 image frames, 120 kVp, 5 mA with a pulsed exposure time of 3.7 s, a voxel size of 0.4 mm and a field of view (FOV) of 16 × 8 or 16 × 11 mm. The MRI scan was taken using a 3.0 T X series Philips Achieva system (Philips Healthcare, Best, The Netherlands) with a dStream Head 32ch coil adopting a 3D_T1W-mDIXON protocol pixel size 0.548781; 336 px width × 336 px height; FOV: 184.39; slice increment 0.5 mm; slice thickness 0.6 mm; total scan time: 279 s.

The CBCT and the MRI scans were saved as DICOM files (Digital Imaging and Communications in Medicine), which are the international standard for transmitting, storing, and processing medical imaging. 

### 2.3. Image Segmentation

Volumetric rendering of DICOM files and segmentation and analysis of the mandibular condylar head were performed using 3D Slicer (open source, version 5.0.2; http://www.slicer.org (accessed on 4 April 2024)) [21] (Figure 1). The 3D Slicer software is similar to a radiology workstation that supports versatile visualizations, but also provides advanced functionality such as automated segmentation and registration for a variety of application domains. Unlike a typical radiology workstation, 3D Slicer is free and not tied to specific hardware. The models were defined by two expert operators (AMB and DC), who conducted the reconstruction of the two condyles for both CBCT and MRI series in triplicates, respectively.

In order to obtain a first orientation of the images from the DICOM files of both the scans, the spatial origin of the original data was aligned to the origin of the global reference system. In each scan, two different thresholds were selected in order to include only bone parts, which were represented by higher and brighter values of the Gray scale for the CBCT and by the lower and darker Gray values for the MRI, respectively. As this study aimed to compare two different techniques in reproducing a selected anatomical portion of the jaw represented by the condyle, a brush selection tool was adopted with the selected thresholds in order to selectively include only the voxels that belonged to the condylar head. This method allowed us to directly obtain models of both condyles of the mandible without adjacent bone portions. The resulting 3D models were then checked in order to correct any sort of error in the reconstruction, and holes inside the geometry were filled, so as to remove undesired structure information. Lastly, the reconstructed volumes for both the scans and for each of the six replicates were exported as STL files.

### 2.4. Model Elaboration

In MATLAB R2022a (https://www.mathworks.com/products/matlab.html (accessed on 8 October 2023)) [22], an original algorithm for semiautomatic best-fit alignment between each model was developed by the authors (Figure 2).

The positions of the first 2 condyles extracted from the CBCT series of the first expert operator were used as a reference model for the alignment. Then, the absor.m function [Matt J (2022). Absolute Orientation—Horn’s method (https://www.mathworks.com/matlabcentral/fileexchange/26186-absolute-orientation-horn-s-method (accessed on 8 October 2023)), MATLAB Central File Exchange] was implemented inside the script to pre-align the other models. This function takes the position of a minimum of three correspondence points for both the reference (REF) and moving (MOV) models as inputs, then returns the transformation matrix as an output by performing the least squares estimation of rotation and translation of the two corresponding point sets. Due to the imprecision of the first alignment in obtaining the best superimposition of the models, corresponding regions of points on both surfaces were selected and used as source nodes for the ICP algorithm, as allowed by the script. Therefore, two points for each model, REF and MOV, were picked in correspondence with the most prominent point in the superior part of the right and left condyles, and a selecting radius from those vertices was set in order to obtain two groups of points, representing the same condyle portion. Then, the MOV-selected point clouds were aligned to the REF ones with an ICP algorithm (pcregistericp.m [23,24,25]), which provided the pose matrix for the alignment of the MOV model to the REF one as an output of the function.

To make comparisons between each model, a standard and previously published protocol was adopted to perform the condylar head cut [26]. All the reoriented meshes were imported in Meshmixer (Autodesk, Inc., San Francisco, CA, USA) [21] (https://www.meshmixer.com/ (accessed on 8 October 2023)) and two orthogonal planes were generated: the former passing through the sigmoid notch points of both condyles and normally aligned to the one of the sigmoid notch node, and the latter being defined by the two sigmoid notch points and with its normal lying on the first plane (Figure 3).

The edges of the condyles were then cut using the two planes as references, and a total of 12 pairs of condyles (6 for MRI and 6 for CBCT), with the lower edges cut according to a previously validated and proven accurate method, were obtained.

### 2.5. Mesh Analysis and Comparison

The refined geometries were then imported into CloudCompare (CloudCompare version 2.20.2, Anoia) [27] (Figure 4), an open-source software for cloud-to-cloud distance computing and mesh analysis. All the condyles were first subdivided into two subsets, the right and left side, then the measures of surface (S) and volume (V) were saved on an Excel workspace.

Because both S and V measures omit all the information regarding the differences in morphology between each condyle of the two sides, a computation of the distances between the respective vertices of all the models was performed with the embedded tool. This operation returns a .csv file histogram with all the intervals of the distance between points on the horizontal axis and the number of vertices per bin on the vertical axis.

### 2.6. Statistical Analysis

The data of S and V obtained from CloudCompare were organized in a Microsoft Excel 2016 spreadsheet (Microsoft Corporation, Redmond, WA, USA) by dividing the data in three condyles per side. The aim was first to assess if there were statistically relevant differences between the resulting models, using a Mann–Whitney–Wilcoxon nonparametric test. Secondly, groups of data coming from both operators were generated, and a Bland–Altmann analysis was performed in Excel in order to assess the reproducibility of the method between all the models of the two experts. Two geometrical parameters were selected, i.e., the total surface (S) and volume (V) of the single condyle model. The aim was to quantify the intraclass and interclass variability from the mean of each DICOM series (CBCT and MRI).

These results were divided into intraclass variability in S and V between CBCT models, intraclass variability in S and V between MRI models, and interclass variability in S and V between MRI and CBCT models.

The mean value was firstly computed for all V and S of the two separated sides and subtracted from all the other values to obtain the *random_error* and *random_error%*:*random_error* = *value-mean_value*
*random_error%* = (*random_error*)/(*mean_value*) × 100

In MATLAB, a Kruskal–Wallis nonparametric test with a Post Hoc Bonferroni correction was performed on distance data coming from the computation in CloudCompare, cleaned from outlier values that exceeded the µ ± 3σ interval. Even in this case, the results were divided into intraclass variability between points of CBCT models, intraclass variability between points of MRI models, and interclass variability between points of MRI and CBCT models.

## 3. Results

### 3.1. Mann–Whitney–Wilcoxon Test

The S and V values of all the models obtained from experts AMB and DC, respectively, were tabulated in Excel. Then, mean values and standard deviations were calculated for each group, divided into S and V values for each imaging sequence of the single operator, as shown in Table 1.

In order to find any statistically significant difference between the S and V values of AMB and DC, a Mann–Whitney–Wilcoxon nonparametric test was performed by comparing two groups: one that compared the values of S_CBCT_ (AMB) and S_MRI_ (AMB) with S_CBCT_ (DC) and S_MRI_ (DC) and the other that compared the values of V_CBCT_ (AMB) and V_MRI_ (AMB) with V_CBCT_ (DC) and V_MRI_ (DC).

The mean values with standard deviation for S and V and the *p*-values related to the comparisons are presented in Table 2. Both resulting *p*-values were higher than the 0.05 significance level, confirming that models coming from the two experts could be considered equivalently comparable to each other in order to continue the assessment of the stability of the procedure.

### 3.2. Bland–Altmann Analysis on Volumes and Surfaces

All the absolute and relative deviation values were plotted in Excel. The numbers on the horizontal axis represent the single condyle, while on the vertical axis, the errors are reported as absolute values and percentages over the mean value (Figure 5, Figure 6 and Figure 7). The numbers represent the values obtained by the operators: those from 1 to 3 regard the former (AMB), while those from 4 to 6 refer to the latter (DC). As depicted in Figure 5, the models obtained from CBCT are very similar to each other, with *random_error%* in V and S not exceeding the value of 3% for both operators. Even considering the differences in µL (absolute value) for V, the SD from the mean (represented by the dashed lines in the charts) was lower than 29.89 µL.

For MRI values, both experts also obtained very similar results for the modeling of the right condyle, with values of SD that were under the threshold of 6 µL. These data confirmed the fact that, for well-defined MRI images, it is possible to extract bony morphology. The results regarding the left condyle are different, since there is an evident increase in the deviations from the mean, with a peak value of 60.99 µL for the second replication of the first operator. In this case, the higher variability of the left side compared to the right one was due to worse image reconstruction during the acquisition time of the MRI sequence. These values are also higher if compared to the contralateral part where none of the reconstructions exceed the 5% error in V and 2% for S (all the values are related to the mean V and S of the six condyles in Figure 6).

Concerning the Bland–Altmann analysis performed on the comparison between MRI and CBCT values, all S and V values of the MRI models were compared to the mean S and V of the CBCT geometries. The latter were taken as gold standard values (Figure 7). In this case, there is an evident underestimation of both S and V values for all the geometries obtained from the 3T-MRI sequence. In particular, the surface area of these models varied from CBCT ones, at least by 9.58% over the mean S value of the gold standard. Also, V values were one-fifth lower than the mean CBCT volume. Even in the presence of an evident shrinkage of the condyles’ geometry in every dimension, it is noticeable that only a small variability in both S and V values was found for all the right models, while a slight increase was found for the left counterpart.

### 3.3. Mesh Comparison

Regarding the evaluation of similarities between the models of the same group (Appendix A), all the geometries were confronted by pairwise comparison to the first ones, arbitrarily selected as references, and the linear deviations between the vertices of the mesh were grouped and plotted in histograms with 60 equivalent bins, fitted with a Gaussian curve to highlight the nonparametric distribution and median and interquartile range values displayed on the top right corner (Appendix A). Tables under the charts (Appendix A) show mean values and standard errors for each error distribution, named as GROUP, on the left, and the resulting *p*-values of the Post Hoc test between groups with Bonferroni correction on the right.

For all the CBCT linear deviations, the median values or bias of the model did not exceed 132 µm, with IQRs that were under 172 µm. The Kruskal–Wallis nonparametric test returned, for both condyles’ error distributions, a *p*-value equal to zero, except for three comparisons. A high reproducibility between models was also reported for the MRI sequence, with a better definition for the right side compared to the left one, having median values of distance to the reference mesh that did not exceed 159 µm (IQR < 254 µm). In this case, the small distributions of values also resulted in a *p*-value < 0.05 for both Kruskal–Wallis and Post Hoc tests.

The results of the pairwise comparison between the 3T-MRI condyles and the reference CBCT model displayed in Figure 8 were plotted with the same template previously described. The comparison between the gold standard model of CBCT and the other six MRI ones shows that for the right condyles, all the deviance distributions did not exceed 408 µm (IQR < 523 µm) of underestimation with respect to the reference and they were not significantly different from each other. This was not confirmed on the left side, where the median underestimation was <390 µm (IQR < 493 µm) and only a few models did not differ from each other in a statistically significant way.

## 4. Discussion 

The purpose of this study was to evaluate whether MRI can serve as an alternative diagnostic and research tool to CBCT in the 3D evaluation of maxillofacial structures [19]. This study, to our knowledge, is the first in this field, meaning it is the first study to utilize both MRI and cone beam CT to assess the volume of the mandibular condylar head and to provide a method for their comparison. To this end, the authors performed, following established protocols, a 3D volumetric assessment of the mandibular condylar head on both MRI and CBCT scans. It was thus possible to assess in vivo the potential of a 3T-MRI scan to obtain meaningful volumetric reconstructions potentially useful for clinical and research purposes, and to compare MRI to the current clinical benchmark under clinically representative conditions. 

The authors showed that the mean values of the CBCT series differed between each operator by only a few mm^2^ of surface area and only 1 mm^3^ of volume. Although CBCT remained the elective technique in terms of robustness and liability, 3T MRI allowed an excellent approximation that could be estimated in a difference between mean values of S and V, for AMB and DC, around an order of magnitude higher than CBCT (mean difference of 0.012 mL over a mean volume of 12 mL, equal to 1% of estimation error over the total mean volume, and about 1.5% of estimation error over the total mean surface). As expected [28], MRI showed a higher standard deviation compared to CBCT, but the values were similar between the operators. The similarity of the SD between different operators makes the authors confident of the probable stability of the proposed method, even in this kind of MRI scan.

The variability in the values reported for the CBCT models was due to the fact that the DICOM sequence used was not perfectly defined in the condylar zone, as a result of a low exposure time setting of the machine. Also, the patient presented partial resorption of the bone in the condylar segment that led to a lower radiopacity and subsequently lower definition of the structure.

Usually, MRI sequences are not used clinically to inspect bone tissues, because the imaging technique cannot extract information relative to the internal trabecular structure. This corresponds to a black and normally poorly resolute representation of the zone [29]. In this case, performing a 3D reconstruction from 3T-MRI sequences allowed a better definition of the bone edges thanks to the better resolution in soft tissue representation [30]. The higher variability reported for models on the left side compared to the right one was attributable to a worse image reconstruction during the acquisition.

Comparing MRI models to CBCT ones, it is primarily evident that all the MRI replicates show an underestimation of both the values of S and V. This happens because the technique itself does not allow a high definition of bone structures to be achieved; therefore, the contours defined by the soft tissues that envelope condyles were used as boundaries by the operators for threshold selection [28]. This assumption, however, carries a series of shortcomings, firstly related to the information that lies at the interface between bone and soft tissues. Indeed, to exclude all the neighboring structures that are not part of the mandible but have similar Gray values on the Hounsfield scale, the experts adopted a common threshold for bone segments that considers only the pixels with small Gray values and differs from those selected for the surrounding soft tissues. However, this type of method, similar to the majority of segmentation algorithms, depends on the quality of the DICOM images processed. This aspect was particularly noticeable in zones such as the one highlighted in Figure 9 (red circle), where the slice of 3T-MRI was partly affected by noise corresponding to the interface between hard and soft tissues. This could cause a loss of information regarding the contours between the two tissues, possibly resulting in an under/overestimation of the cortical bone thickness. 

Another factor that affected the estimation of the condyle volume was that tissues with small amounts of water inside their matrix, like calcified ones, do not transmit energy to the MRI detector. As a consequence, the result is a darker representation of that specific anatomical part where only information about the external boundaries can be extracted by selectively excluding soft tissue portions with fine thresholding.

A possible way to overcome some of these issues could be considering MRI sequences obtained with a 3T machine set with a prolonged exposure time [1]; this way, the images could be much clearer than those generated with a smaller permanent magnet.

Regarding the mesh comparison, the authors aimed to test whether there was reproducibility between all the reconstructed models by the operators with both MRI and CBCT scans by assessing the linear discrepancies between models obtained from each scanning method. As expected, all the linear deviations obtained between CBCT models were lower than those measured for the MRI group. As a result, small distributions of linear deviations between models, which differed from each other in a statistically relevant way, were found; this was mainly due to the fact that few µm of linear difference can be statistically relevant when comparing small distributions, as shown in the table of Figure 8.

Since the main purpose of this work was to test the reliability of the proposed method that aims to use MRI sequences instead of CBCT to clinically evaluate condylar growth in young people, the final comparison between the gold standard model of CBCT and the other six MRI ones shows that for the right condyles, all the deviance distributions, except one, were not significantly different from each other. This was not confirmed on the left side because of the previously reported problems with image resolution.

Even if there was a systematic underestimation of the geometrical dimensions for the anatomies obtained from 3T-MRI compared to the gold standard, it is evident that this new method boasts high reproducibility through different times and operators. The authors believe that it could be a reliable and stable method to compare anatomical structures pre- and post-treatment.

### Strengths, Limitations, and Future Perspectives

In recent years, various protocols for volumetric MRI scanning have garnered attention in the fields of orthopedics, maxillofacial surgery, and dentistry, aiming to serve as an alternative to the reference standard of 3D volumetric analysis, namely cone beam CT. Several types of sequences are currently being tested, including black bone imaging, ultrashort/zero echo time (UTE/ZTE) sequences, and T1-weighted 3D gradient-echo sequences [1,31].

It may be worth mentioning some disadvantages of MR bone imaging. Compared with CT, the scanning time for MR imaging, including MR bone imaging, is long. The addition of MR bone imaging to conventional sequences would further extend the total scanning time, possibly being a drawback, especially in children. Another disadvantage is represented by susceptibility artifacts, which are usually seen in the presence of para/ferromagnetic substances or structures [19]. 

Nevertheless, in this work, the segmentation of MRI scans ensured great consistency between different operators and between different segmentations taken by the same operator. These characteristics could prove favorable for studying the three-dimensional morphology of the TMJ without the burden of X-ray exposure that is inherently connected to CBCT scans. In fact, the main limitation of the use of CBCT in longitudinal studies is its biological cost that, however, does not apply to MRI. Considering the overall good concordance with CBCT (the gold standard for bone measurements) and the absence of radiation exposure, 3T-MRI bone assessment could be performed in the future to reduce radiation dose, which is crucial in young patients, and overcome the limitations of conventional bidimensional imaging. It could be a useful resource in assessing soft and hard tissues of the facial area for clinical and research purposes, i.e., surgery and implant planning, growth study in orthodontics, and even orthognathic surgery planning.

The development of MRI scans could give us answers to several orthodontic issues on what really happens in the TMJ in response to different orthodontic and dental interventions and will help in the 3D analysis and treatment of the TMJ. In the near future, the capability of MRI sequences to perform volumetric analysis of soft and hard tissues will be of paramount relevance in the field of regenerative medicine to study, test, and develop patient-tailored 3D-printed grafts.

## 5. Conclusions

Despite all the actual limitations of MRI scans related to their capability of representing bone tissues and the major possibility of having motion artifacts during the acquisition time, the tested method appears to be a viable option for the volumetric analysis of mandibular condyles, worthy of being tested on a larger pool of data. In fact, the difference between MRI and CBCT shown in the Bland–Altmann analysis highlighted an error between the two methods that appears to be systematic and allegedly due to resolution issues that are inherent to the imaging modality. This could be overcome through applying a corrective coefficient that could be developed by replicating this method on a larger pool of scans.

## Figures and Tables

**Figure 1 dentistry-12-00220-f001:**
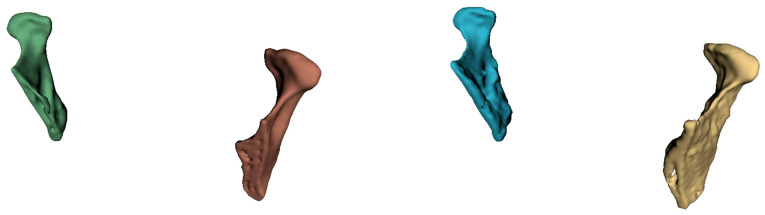
Model rendering in 3D slicer: representation of reconstructions obtained from CBCT (**left**) and MRI images (**right**).

**Figure 2 dentistry-12-00220-f002:**
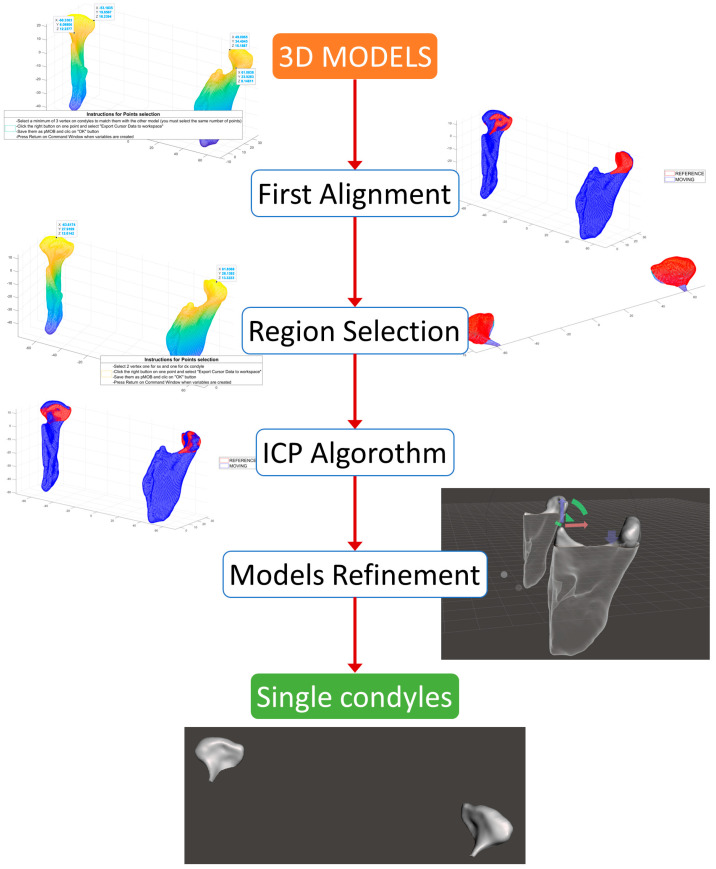
Workflow of the proposed method, from the 3D models obtained in 3D Slicer to the single condyles used for mesh analysis(the models to be oriented are represented in rainbow and blue colors, while the reference one is highlighted in red).

**Figure 3 dentistry-12-00220-f003:**
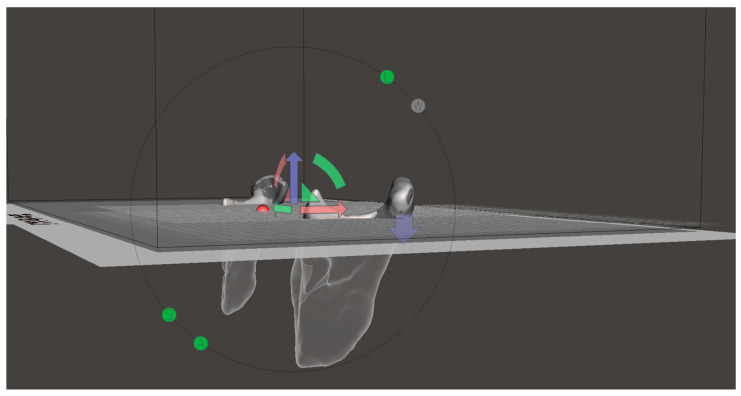
Mesh-cut operation in Meshmixer with the cutting plane (passing through Anterior Nasal Spine and L/R Sigmoid Notch) highlighted (arrows in red, green and blue represents the reference triad of the cutting plane).

**Figure 4 dentistry-12-00220-f004:**
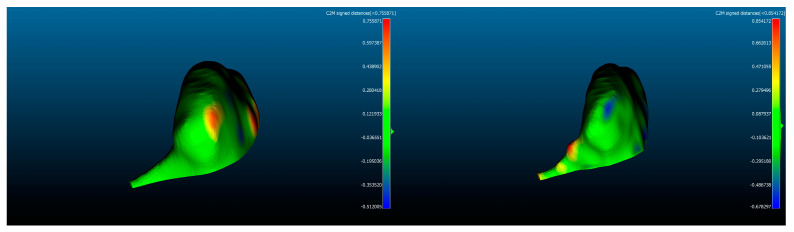
Heatmaps resulting from mesh comparison in CloudCompare: CBCT (**left**) and MRI (**right**) models. The blue and red regions represent the areas with the maximum discrepancies from the reference model.

**Figure 5 dentistry-12-00220-f005:**
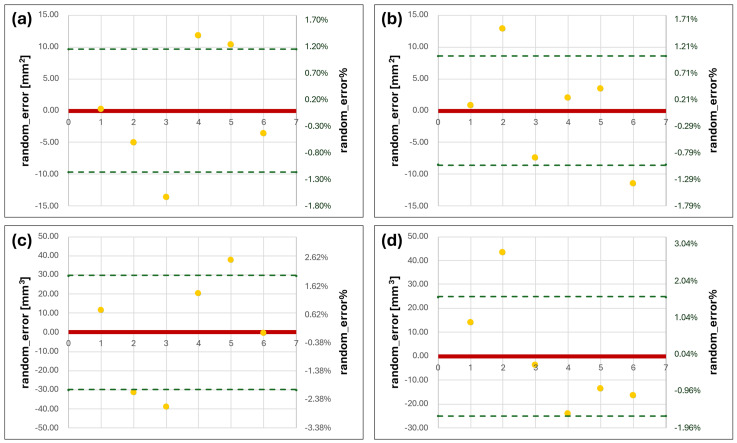
Bland–Altmann charts for V and S of CBCT series. (**a**,**b**) are the errors in S evaluated for right and left condyles, respectively; (**c**,**d**) are the errors for the right and left V values (yellow dots correspond to the S or V random_error for each condyle model from the mean, the solid red line represent the zero level of the vertical axis and the green dashed lines correspond to the positive and negative standard deviation values from the mean).

**Figure 6 dentistry-12-00220-f006:**
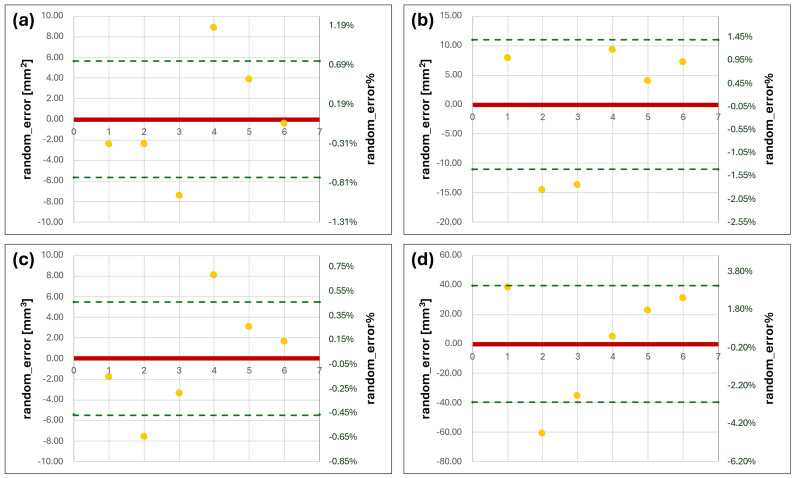
Bland–Altmann charts for V and S of MRI series. (**a**,**b**) are the errors in S evaluated for right and left condyles, respectively; (**c**,**d**) are the errors for the right and left V values (yellow dots correspond to the S or V random_error for each condyle model from the mean, the solid red line represent the zero level of the vertical axis and the green dashed lines correspond to the positive and negative standard deviation values from the mean).

**Figure 7 dentistry-12-00220-f007:**
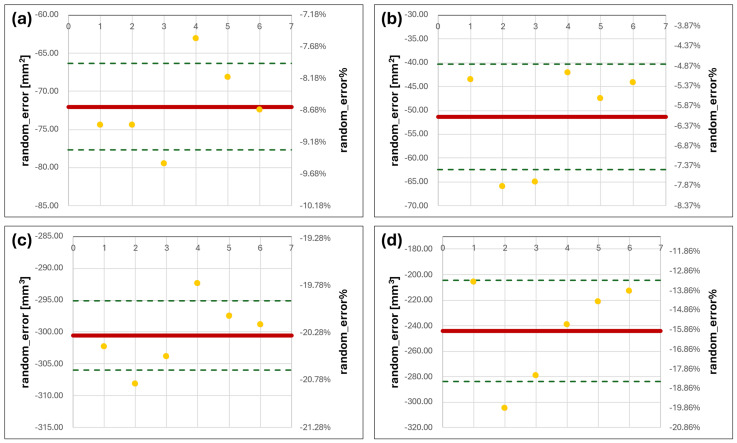
Bland–Altmann charts for S and V of MRI series compared to CBCT values. (**a**,**b**) are the errors in S evaluated for right and left condyles, respectively; (**c**,**d**) are the errors for the right and left condyles’ V values (yellow dots correspond to the S or V random_error for each condyle model from the mean CBCT one, the solid red line represent the random_error mean level and the green dashed lines correspond to the positive and negative standard deviation values from the mean).

**Figure 8 dentistry-12-00220-f008:**
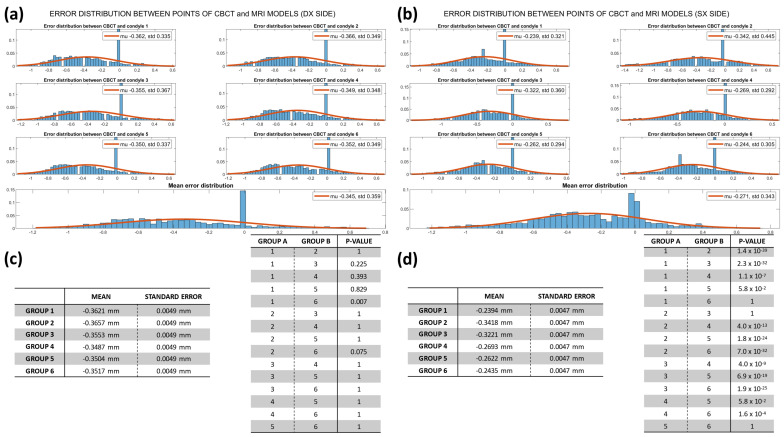
Distribution of deviation computed in CloudCompare between CBCT model 1 and the other six MRI models for each side. (**a**,**b**) Error distribution for the right and left sides, respectively, with the mean error distribution plotted at the bottom. (**c**,**d**) Tables with mean deviations and standard errors and with resulting *p*-values of the Post Hoc Bonferroni test.

**Figure 9 dentistry-12-00220-f009:**
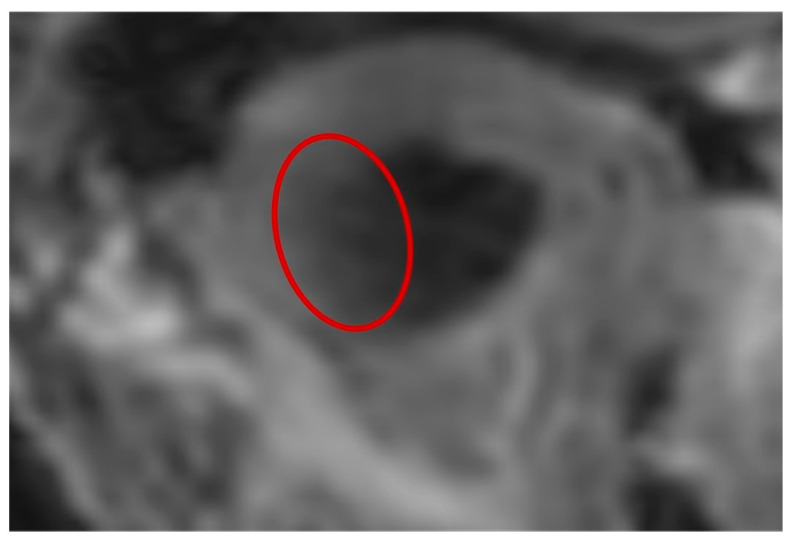
Detail of condyle edges corresponding to the temporal zone. The red circle highlights an area on the edge between the hard and soft tissue of the condyle head, which is not well defined because it is affected by noise.

**Table 1 dentistry-12-00220-t001:** Mean and standard deviation values of CBCT and MRI models for both the experts, abbreviated as AMB and DC.

Values	No. of Samples	Mean	SD
SCBCT (AMB)	6	833.69 mm^2^	9.33 mm^2^
VCBCT (AMB)	6	1506.34 mm^3^	55.12 mm^3^
SCBCT (DC)	6	837.85 mm^2^	8.43 mm^2^
VCBCT (DC)	6	1507.73 mm^3^	15.53 mm^3^
SMRI (AMB)	6	768.62 mm^2^	13.29 mm^2^
VMRI (AMB)	6	1222.89 mm^3^	61.56 mm^3^
SMRI (DC)	6	779.50 mm^2^	13.81 mm^2^
VMRI (DC)	6	1246.56 mm^3^	69.46 mm^3^

**Table 2 dentistry-12-00220-t002:** Mean and standard deviation values of CBCT and MRI models for both the experts (AMB and DC).

	AMB (Mean ± SD)	DC (Mean ± SD)	*p*-Value
S (CBCT + MRI; *n* = 12)	801.16 ± 35.7 mm^2^	808.68 ± 32.36 mm^2^	0.2013
V (CBCT + MRI; *n* = 12)	1364.62 ± 158.16 mm^3^	1377.15 ± 144.59 mm^3^	0.3975

## Data Availability

Data are available upon reasonable request from the first author’s email address.

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
