# Peer review of "Assessing the Volume of the Head of the Mandibular Condyle Using 3T-MRI—A Preliminary Trial"

_dentistry, 2024, doi:10.3390/dj12070220_

Round 1

Reviewer 1 Report

Comments and Suggestions for Authors

The authors extracted mandibular condyles from 3D images of the mandible using 3T-MRI as a reference to CBCT and presented the results of their measurements of area and volume. While I have no objections to the purpose of the study or the methodology employed, there are some statements in the results and conclusions with which I disagree. My comments are as follows:

 1. The debate over whether bone tissue can be evaluated using MRI is currently a hot topic and an intriguing point of discussion. While the authors attempted to extract mandibular condyles from 3D images using T1WI, recently, imaging protocols similar to CT images have emerged, and it is essential to validate their measurement accuracy. If MRI can provide sufficient bone evaluation of the temporomandibular joint (TMJ), it will be necessary to consider how to utilize it clinically in the future. I would like to request the authors to add comments on this aspect as future prospects in the Discussion section.

2. While it is significant that the measurement error is sufficiently small, there is a considerable difference in the actual measured volumes and areas of CBCT and MRI as seen in Table 1. While this aspect is addressed in the Discussion, the basis for stating in the Conclusion that "In fact, the difference between MRI and CBCT appears to be of little or no clinical relevance" is unclear. This statement is somewhat emphasized and presents a point of disagreement.

3. Related to Comment 2 above, there is a lack of Limitations mentioned in this study. As far as I can see, the small sample size and the underestimation of volumes and areas due to the signal void of MRI bone tissue should be noted in this section. I agree that points for potential resolution in the future are valid.

4. It might be helpful to add abbreviations to make it clearer that AMB and DC in Tables 1 and 2 refer to the initials of the observers.

5. It's not clear what Figure 11 is depicting. Consider either updating it with a clear image or adding a detailed explanation.

Author Response

All changes in the text of the paper have been highlighted in yellow to make them more easily identifiable. For your convenience, our response has been provided in bold text below each of your comments below

  1. The debate over whether bone tissue can be evaluated using MRI is currently a hot topic and an intriguing point of discussion. While the authors attempted to extract mandibular condyles from 3D images using T1WI, recently, imaging protocols similar to CT images have emerged, and it is essential to validate their measurement accuracy. If MRI can provide sufficient bone evaluation of the temporomandibular joint (TMJ), it will be necessary to consider how to utilize it clinically in the future. I would like to request the authors to add comments on this aspect as future prospects in the Discussion section.

RESPONSE: Thank you for your precious comments. We actually feel that they were able to improve significantly our manuscript in delivering our messages. We implemented the discussion section according to your suggestion. We introduced a whole paragraph regarding streghts, limitations and future development at the end of discussions (4.1).

  1. While it is significant that the measurement error is sufficiently small, there is a considerable difference in the actual measured volumes and areas of CBCT and MRI as seen in Table 1. While this aspect is addressed in the Discussion, the basis for stating in the Conclusion that "In fact, the difference between MRI and CBCT appears to be of little or no clinical relevance" is unclear. This statement is somewhat emphasized and presents a point of disagreement.

RESPONSE: Thank you for your observation. We made it clearer into the conclusions our message and removed the misleading sentence

  1. Related to Comment 2 above, there is a lack of Limitations mentioned in this study. As far as I can see, the small sample size and the underestimation of volumes and areas due to the signal void of MRI bone tissue should be noted in this section. I agree that points for potential resolution in the future are valid.

RESPONSE: We agree with your comment. We implemented discussion and conclusion according to your suggestion

  1. It might be helpful to add abbreviations to make it clearer that AMB and DC in Tables 1 and 2 refer to the initials of the observers.

RESPONSE: We changed the legend of table 1 according to your suggestion

  1. It's not clear what Figure 11 is depicting. Consider either updating it with a clear image or adding a detailed explanation.

RESPONSE:Thank you for your suggestion. We improved the legend of figure 11 according to your suggestion

Reviewer 2 Report

Comments and Suggestions for Authors

I would like to express my appreciation to the authors for their submission of the article titled "Assessing the Volume of the Head of the Mandibular Condyle Using 3T-MRI: A Proof of Concept" to the Dentistry Journal. The topic addressed in the paper is undoubtedly intriguing and holds significant relevance in the field of craniofacial imaging. However, upon careful review, it is evident that the article requires major revisions before it can be considered for publication. Below, I have outlined several areas that need attention and improvement for the manuscript to meet the standards of the journal.

Introduction:

The introduction provides a well-written overview of the topic; however, the emphasis on cephalometric analysis may not be directly relevant to the subject matter discussed in the article. Instead, considering the novelty and significance of the study, I recommend focusing the introduction on the potential of non-ionizing alternatives for three-dimensional assessment of maxillofacial structures, particularly highlighting the role of 3T-MRI in this context. This shift in focus would better align the introduction with the core objective of exploring the feasibility of 3T-MRI in assessing the volume of the mandibular condyle head.

Materials and Methods:

The Materials and Methods section requires substantial revision. It would benefit from a more organized structure, perhaps drawing inspiration from other articles such as the one referenced (DOI: 10.1111/cid.13160). Avoiding bulleted lists and opting for a narrative format would enhance the clarity and flow of the description. By providing a coherent and detailed account of the study procedures, the revised Materials and Methods section would facilitate replication and interpretation of the findings by the readership.

Results:

The Results section should present findings in a clear and concise manner, avoiding subdivision into sub-paragraphs. Some parts of the text included in the Results appear to pertain more to the Materials and Methods section. Ensuring a seamless transition between the presentation of results and the description of methodology would enhance the readability and coherence of the manuscript. Additionally, focusing on key findings and providing necessary context without excessive detail would improve the overall impact of the Results section.

Discussion:

The Discussion section requires a complete rewrite to align with the suggested structure. It should be organized as follows:

(a) Statement of Principal Findings: Begin by succinctly summarizing the main findings of the study.

(b) Strengths and Weaknesses of the Study: Evaluate the strengths and limitations of the research methodology employed, acknowledging any potential biases or confounding factors.

(c) Strengths and Weaknesses in Relation to Other Studies: Compare and contrast the findings of the current study with those of previous research, highlighting any discrepancies or consistencies.

(d) Meaning of the Study: Possible Mechanisms and Implications: Interpret the implications of the study results, discussing potential mechanisms underlying the observed outcomes and their relevance to clinical practice or theoretical frameworks.

(e) Unanswered Questions and Future Research: Identify any remaining uncertainties or areas requiring further investigation, proposing avenues for future research to address these gaps.

Revising the Discussion section in accordance with this structure will enhance the clarity and coherence of the manuscript.

Conclusions:

The Conclusions section is generally well-written but could benefit from increased conciseness. While summarizing the key findings and implications of the study, strive for brevity to ensure that the main takeaways are effectively communicated to the reader.

I would like to express my appreciation for the opportunity to review this manuscript. I eagerly await the revised version, incorporating the suggested revisions, and I look forward to seeing the final outcome of this research.

Sincerely,

Comments on the Quality of English Language

As English is not my native language, I prefer to refrain from assessing the quality of English in the article. However, I highly recommend having the text proofread by a native English speaker or a professional editor. Thorough proofreading can help ensure clarity, consistency, and fluidity of the text, thereby enhancing overall communication of the topics addressed in the article.

Author Response

All changes in the text of the paper have been highlighted in yellow to make them more easily identifiable. For your convenience, our response has been provided in bold text below each of your comments below

1. Introduction:

The introduction provides a well-written overview of the topic; however, the emphasis on cephalometric analysis may not be directly relevant to the subject matter discussed in the article. Instead, considering the novelty and significance of the study, I recommend focusing the introduction on the potential of non-ionizing alternatives for three-dimensional assessment of maxillofacial structures, particularly highlighting the role of 3T-MRI in this context. This shift in focus would better align the introduction with the core objective of exploring the feasibility of 3T-MRI in assessing the volume of the mandibular condyle head.

RESPONSE: Dear reviewer,

We appreciate your observation, and we believe it has helped improve the fluency of the text and the logical sequence of topics covered in the introduction. Indeed, we have modified the opening of the introductory section of the article while also retaining a portion of our original introduction.

The purpose of our proof of concept is to test the feasibility of performing a morphological study of the mandibular condyle. But why do we need to do this? We aim to better understand the still obscure mechanisms of growth and adaptation, which we hope to shed light on with this technique. It will be possible to overcome the limitation inherent in CT scans, namely ionizing radiation, and move beyond the two-dimensionality of cephalometric tracings.

Furthermore, this study aims to provide a method based on open-source software or widely available softwares (i.e., MATLAB) to standardize future comparative studies between CT and MRI. For this reason, in addition to the introduction suggested by you, we have decided to retain part of the previous introduction.

Thank you!

2. Materials and Methods:

The Materials and Methods section requires substantial revision. It would benefit from a more organized structure, perhaps drawing inspiration from other articles such as the one referenced (DOI: 10.1111/cid.13160). Avoiding bulleted lists and opting for a narrative format would enhance the clarity and flow of the description. By providing a coherent and detailed account of the study procedures, the revised Materials and Methods section would facilitate replication and interpretation of the findings by the readership.

RESPONSE: We agree with your suggestion. We have therefore decided to implement the text in accordance with your suggestions.

3. Results:

The Results section should present findings in a clear and concise manner, avoiding subdivision into sub-paragraphs. Some parts of the text included in the Results appear to pertain more to the Materials and Methods section. Ensuring a seamless transition between the presentation of results and the description of methodology would enhance the readability and coherence of the manuscript. Additionally, focusing on key findings and providing necessary context without excessive detail would improve the overall impact of the Results section.

RESPONSE:

We partially agree with this comment. The comment has been taken into consideration and partly implemented in the text, with certain benefits to the readability of the manuscript, and for this, we thank the reviewer. Given the numerous statistical tests conducted, and considering that understanding the organization of data in each statistical test is essential to understanding the test itself, paragraph subdivision is necessary to ensure this section of the article is orderly and comprehensible. At present, thanks to your observations, there doesn't appear to be any misplaced text in the two sections you mentioned, and the subdivision of results into paragraphs seems to enhance their comprehensibility for the reader.

4. Discussion:

The Discussion section requires a complete rewrite to align with the suggested structure. It should be organized as follows:

(a) Statement of Principal Findings: Begin by succinctly summarizing the main findings of the study.

(b) Strengths and Weaknesses of the Study: Evaluate the strengths and limitations of the research methodology employed, acknowledging any potential biases or confounding factors.

(c) Strengths and Weaknesses in Relation to Other Studies: Compare and contrast the findings of the current study with those of previous research, highlighting any discrepancies or consistencies.

(d) Meaning of the Study: Possible Mechanisms and Implications: Interpret the implications of the study results, discussing potential mechanisms underlying the observed outcomes and their relevance to clinical practice or theoretical frameworks.

(e) Unanswered Questions and Future Research: Identify any remaining uncertainties or areas requiring further investigation, proposing avenues for future research to address these gaps.

Revising the Discussion section in accordance with this structure will enhance the clarity and coherence of the manuscript.

RESPONSE : Dear reviewer, the structure you have provided is certainly appropriate and has been partly implemented in our text to meet the journal's requirements. However we are please to let you know that our study, to our knowledge, is the first in this field, meaning it is the first study to utilize both MRI and cone beam CT to assess the volume of the mandibular condylar head and to provide a method for their comparison. Therefore, we have added a paragraph in the discussion section focusing on the strengths and weaknesses of the study and potential future developments, also concentrating on possible clinical implications. As our study is the first of its kind, comparing it with similar articles in dentistry is not feasible, so we have focused on describing and commenting on the numerous results and discussing the strengths and weaknesses of this study.

5.Conclusions:

The Conclusions section is generally well-written but could benefit from increased conciseness. While summarizing the key findings and implications of the study, strive for brevity to ensure that the main takeaways are effectively communicated to the reader.

RESPONSE: We summarized conclusion according to your suggestion. Thank you

Reviewer 3 Report

Comments and Suggestions for Authors

Dear Authors,

your work is exciting. The topic is innovative, current, and useful for both the scientific and academic world. However, I believe it is extremely limiting to have to carry out evaluations and analyzes based on just one case. Arriving at conclusions using this method is neither appropriate nor correct. In light of what has been written, my invitation is to increase the sample also under an evaluation of the sample size and create a work that can become a milestone in this area

Comments on the Quality of English Language

 Moderate editing of English language required

Author Response

Dear Authors,

your work is exciting. The topic is innovative, current, and useful for both the scientific and academic world. However, I believe it is extremely limiting to have to carry out evaluations and analyzes based on just one case. Arriving at conclusions using this method is neither appropriate nor correct. In light of what has been written, my invitation is to increase the sample also under an evaluation of the sample size and create a work that can become a milestone in this area

Thank you for your suggestion. We appreciate your words of praise. A study on multiple subjects has just been sent for evaluation to the ethic committee. However, this article serves as a proof of concept, and its purpose is different. The aim of this article, much like the publication of a protocol for a systematic review, is to test a method for segmenting a specific anatomical part (namely the mandibular condylar head) using two different types of 3D imaging (i.e. MRI and CBCT) and comparing it in a reproducible manner, with the intention of studying it in a future study with an adequate sample size. Essentially, this scientific work lays the foundation for the subsequent study you. Thank you for your support!

Round 2

Reviewer 1 Report

Comments and Suggestions for Authors

The authors provided appropriate explanations and revisions in response to my comments. Therefore, I have no further issues to raise.

Author Response

Dear reviewer, we are glad to know the revised version of our paper found you satisfied and we wish to thank you for your valuable comments that helped to improve our manuscript.

Reviewer 2 Report

Comments and Suggestions for Authors

Thank you for your diligent revisions to the manuscript. Overall, the improvements are commendable and have strengthened the quality of the paper.

We appreciate your efforts in addressing the comments raised during the review process. However, we kindly suggest considering moving Figures 8 and 9 to the supplementary materials to streamline the main text. Additionally, removing the formula on page 8 would enhance readability without sacrificing the integrity of the content.

Furthermore, it would be beneficial to include references in the discussions section to support your arguments and strengthen the scientific perspective.

Once again, we commend you on the progress made and look forward to the final version of the manuscript.

Author Response

REVIEWER: Thank you for your diligent revisions to the manuscript. Overall, the improvements are commendable and have strengthened the quality of the paper.
We appreciate your efforts in addressing the comments raised during the review process.

AUTHORS' RESPONSE: Dear reviewer, thank you for your help in improving our manuscript. We implemented the text with your suggestions. 

However, we kindly suggest considering moving Figures 8 and 9 to the supplementary materials to streamline the main text. Additionally, removing the formula on page 8 would enhance readability without sacrificing the integrity of the content. Furthermore, it would be beneficial to include references in the discussions section to support your arguments and strengthen the scientific perspective. Once again, we commend you on the progress made and look forward to the final version of the manuscript.
AUTHORS' RESPONSE: Dear reviewer, thank you for your further suggestions. We moved Figures 8 and 9 to the supplementary material – there named Figure S1 and S2, respectively. 

In accordance to your suggestion we removed the formula at page 8.

Further references have also been added in the discussion section (references 28 to 33) 

Reviewer 3 Report

Comments and Suggestions for Authors

Dear Authors,

The objective of your paper and the intent you have set are obvious to me. However, the work thus designated cannot be published in a magazine such as Dentistry Journal. I advise you to choose a less ambitious path

Comments on the Quality of English Language

 Moderate editing of the English language required

Author Response

Dear reviewer, although we appreciate your suggestions, we humbly disagree. With all the limitations of a single-case proof of concept being considered, we hope that this work can represent the first step, deserving to be published, in a series of more structured and powerful studies.

Round 3

Reviewer 3 Report

Comments and Suggestions for Authors

I'm sorry but I stand by my opinion. the article is not publishable

Comments on the Quality of English Language

 Extensive editing of the English language required

Author Response

We acknowledge your opinion and take it into consideration.

For future reference, we kindly request that such judgments be expressed in the first round of review and that you provide detailed reasoning for your opinions, which are currently unsupported by any specific justification.

This would enable the editor to make a decision based on actual deficiencies, which you have not highlighted so far, and which neither we nor the other peer reviewers have found.

Had you clearly stated your reasons from the beginning, we could have responded promptly and avoided an additional round of reviews in which we cannot make any substantive changes to the text but can only provide this response.

You have never explicitly stated any specific reasons for the non-acceptance of the work, other than a personal lack of appreciation, which is neither further explained nor supported by arguments we can address.

Therefore, we agree to disagree with you on this point.

We also note that in this last round of review you have indicated that "Extensive editing of the English language is required." Why did you previously indicate that only "Moderate editing of the English language is required" in earlier reviews? Why has your assessment of the quality of our English deteriorated since the initial rounds of review? We inform you, and also the editors, that one of our co-authors has completed a part of his PhD studies in the United States and that the text has been reviewed by him as well as by a colleague who is a proficient English-speaking dentist. Furthermore, none of the other reviewers have noted this deficiency.

Therefore, we question whether your assessment is accurate or if it is somehow unethical to indicate a worsening of the English language after careful revision and acceptance by the other reviewers, none of whom have mentioned this issue. 

Sincerely, The authors